# AutoTriton: Automatic Triton Programming with Reinforcement Learning in LLMs

## Abstract

Kernel development in deep learning requires optimizing computational units across hardware while balancing memory management, parallelism, and hardware-specific optimizations through extensive empirical tuning. Although domain-specific languages such as Triton simplify GPU programming by abstracting low-level details, developers must still manually tune critical parameters such as tile sizes and memory access patterns through iterative experimentation, creating substantial barriers to optimal performance and wider adoption. In this work, we introduce AutoTriton, the first model dedicated to Triton programming powered by reinforcement learning (RL). AutoTriton performs supervised fine-tuning (SFT) to be equipped with essential Triton programming expertise using a high-quality data gathering pipeline, and conducts RL with Group Relative Policy Optimization (GRPO) algorithm, combining a rule-based reward and an execution-based reward to further improve Triton programming ability sequentially. Experiments across five evaluation channels of TritonBench and KernelBench illustrate that our 8B model AutoTriton achieves performance comparable to mainstream large models, including GPT-5 and DeepSeek-R1-0528. Further experimental analysis demonstrates the crucial role of each module within AutoTriton, including the SFT stage, the RL stage, and the reward design strategy. These findings underscore the promise of RL for automatically generating high-performance kernels, and since high-performance kernels are core components of AI systems, this breakthrough establishes an important foundation for building more efficient AI systems. The model and code will be available on Github.

## 1 Introduction

Efficient kernel engineering serves as the foundation for high-performance deep learning systems, enabling models to execute optimally in an increasingly heterogeneous hardware landscape (Abadi et al., 2016; Paszke et al., 2019). Historically, crafting such kernels in GPU programming languages as CUDA has been the exclusive domain of performance engineers, demanding intimate knowledge of hardware architecture and complex parallel programming patterns (Tillet et al., 2019). The advent of Pythonic GPU programming frameworks, most notably Triton (Tillet et al., 2019), has marked a significant leap in programmability. Notwithstanding these advances, such high-level abstractions have not fully eliminated the complexities of performance tuning. Developers are still burdened with the manual configuration of crucial parameters like tiling configurations and data layouts, a process of empirical trial-and-error that represents a primary bottleneck to realizing performance portability and widespread adoption.

Current research in AI-assisted kernel generation has attracted increasing attention. Several benchmarks, such as TritonBench (Li et al., 2025) and KernelBench (Ouyang et al., 2025), have been introduced to systematically evaluate the abilities of LLMs in generating high-performance kernels. In addition to benchmarks, recent work such as AI CUDA Engineer (Lange et al., 2025) has gained widespread interest. This framework leverages general-purpose LLMs as foundation components to construct an automated workflow. However, its adaptability and flexibility remain limited due to the inherent capability boundaries of the underlying models.

In this work, we introduce AutoTriton, the first model dedicated to Triton programming powered by reinforcement learning (RL). AutoTriton is built upon Seed-Coder-8B-Reasoning (Zhang et al.,

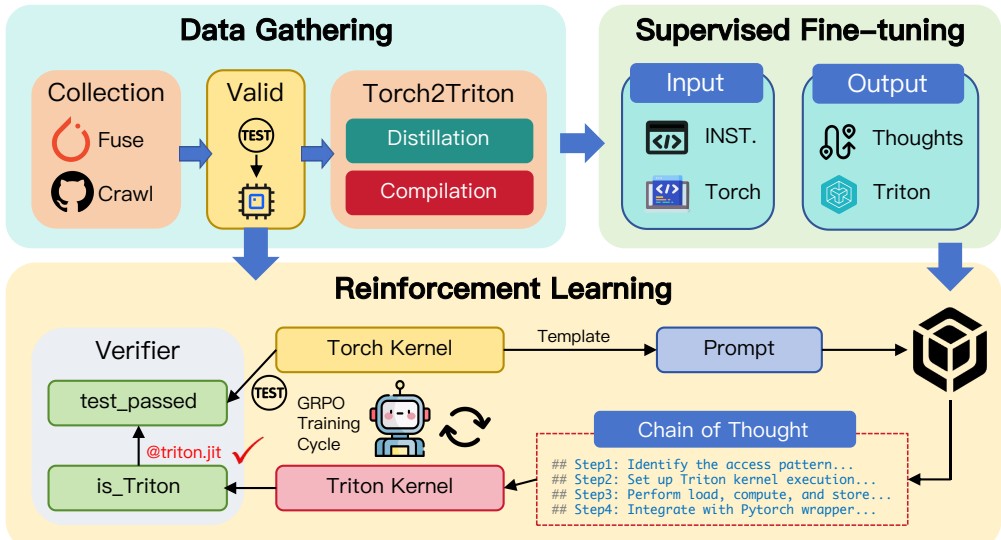

Figure 1: Overview of AUTOTRITON pipeline. The entire pipeline consists of three components: data collection, SFT stage, and RL stage.

2025), which is a reasoning model dedicated to programming, further enhanced through a synergistic combination of supervised fine-tuning (SFT) and RL, which is shown in Figure 1. In the SFT phase, we first design and implement a dedicated data construction pipeline. This pipeline is instrumental in assembling a high-quality Triton dataset that explicitly elucidates key programming concepts and reasoning steps inherent to Triton, thereby equipping AUTOTRITON with foundational programming capabilities. Subsequently, we leverage the data generated from the pipeline again and conduct RL with a combined rule-based and execution-based reward. This phase encourages the model to explore and internalize effective Triton programming strategies, allowing it to capture practical nuances and efficiencies that are challenging to instill through supervised fine-tuning alone.

Experimental results on two representative benchmarks TRITONBENCH and KERNELBENCH show that AUTOTRITON achieves performance comparable to mainstream LLMs, including GPT-5 (OpenAI, 2025), Claude-4-Sonnet (Anthropic, 2025), Qwen3-32B (Yang et al., 2025), DeepSeek-V3.1 (DeepSeek-AI, 2024), DeepSeek-R1-0528 (DeepSeek-AI, 2025), on all five benchmark channels with only 8B parameters, which indicates the effectiveness of AUTOTRITON in the Triton programming task and highlights the crucial impact of our proposed data gathering pipeline and RL training strategy. Further analysis underscores the pivotal roles of the SFT, RL, and reward design components in AUTOTRITON. These findings offer what we consider to be crucial guidance for future research in this direction.

## 2 RELATED WORK

### 2.1 LLM FOR KERNEL GENERATION

Computation kernel generation is crucial for optimizing AI workloads on diverse hardware. Typical approaches, including MLIR (LLVM Project, 2019), TVM (Apache Software Foundation, 2018), collectively enhance the performance and portability of the AI model, addressing the complexities of modern heterogeneous computing environments (Al-Dujaili et al., 2024). Recently, the automation of GPU kernel generation, critical for optimizing machine learning performance, has attracted significant research attention. The systematic evaluation of LLMs in this domain is facilitated by benchmarks such as KERNELBENCH (Ouyang et al., 2025), which assess the generation of fast and correct kernels in various workloads using metrics like "fast_p". Although frontier models excel at general programming tasks, they often fall short on kernel generation tasks, underscoring the gap between general coding capabilities and specialized kernel optimization demands. Similarly, TRITONBENCH (Li et al., 2025) highlights the challenges LLMs face with domain-specific languages like Triton,

revealing difficulties in generating efficient kernels due to unfamiliarity with Triton's specifications and GPU programming intricacies.

Beyond benchmarks, frameworks like AI CUDA Engineer (Lange et al., 2025) utilize agentic approaches, leveraging LLMs for PyTorch-to-CUDA translation and iterative optimization. Despite achieving notable speedups, these training-free approaches are fundamentally constrained by the inherent limitations of the foundational LLMs. To directly enhance model capabilities, Kevin-32B (Baronio et al., 2025) employs multi-turn RL, enabling the model to learn from environmental feedback and significantly improve kernel correctness and performance through self-refinement, particularly on complex tasks. Furthermore, the DeepSeek-R1 model, augmented with test-time scaling, demonstrates the efficacy of allocating increased inference compute for iterative refinement and verification, achieving high accuracy in KERNELBENCH tasks (NVIDIA Developer Blog, 2025). These advancements collectively indicate a trend towards iterative, feedback-driven methodologies to enhance LLM proficiency in specialized high-performance code generation. Additionally, KERNEL-LLM (Fisches et al., 2025) generates Triton kernels via supervised fine-tuning. Despite achieving reasonable performance, it is fundamentally constrained by the ceiling of imitation learning. It does not leverage exploration, limiting its ability to produce higher-quality Triton kernels. Unlike previous works, in this work, we propose AUTOTRITON, the first model specifically designed for Triton programming with reinforcement learning, which achieves remarkable improvements in five typical benchmark channels.

## 2.2 RL FOR CODE

RL provides a powerful paradigm for agents to learn optimal policies through interaction with dynamic environments, maximizing cumulative rewards. Early applications in code generation formulate the problem within a Markov Decision Process (MDP), where partial programs constitute states and grammar productions serve as actions (Chen et al., 2020). This formulation highlights RL's flexibility in adapting to different levels of abstraction. Modern advancements leverage LLMs, treating the code-generating model as an actor and code generation as actions, with functional correctness derived from unit test results providing the reward signal (Le et al., 2022). This approach has enabled systems such as AlphaCode to achieve competitive performance in complex coding tasks (Li et al., 2022). Similarly, Wei et al. (Wei et al., 2025) explore RL for low-level code generation, where LLMs are trained to produce optimized assembly programs. Their work demonstrates that reinforcement signals tied to execution efficiency can substantially improve code performance, extending RL-based code generation beyond functional correctness. Beyond generation, RL is extensively applied in code optimization, notably for learning optimal sequences of compiler passes (Bendib et al., 2024; Shahzad et al., 2022). Here, states are often represented by intermediate representation (IR) statistical analyses or graph-based models, and rewards are tied to performance metrics such as cycle count, area, or resource utilization (Shahzad et al., 2022). The success of frameworks such as CYCLE further demonstrates RL's capacity for iterative self-refinement of faulty code generations, learning from execution feedback, and significantly improving refinement capabilities (Ding et al., 2024).

Despite these advancements, RL for code faces substantial challenges. The design of robust reward functions remains a primary concern. Poorly engineered rewards can lead to unintended behaviors or "reward hacking", where the agent exploits the reward structure rather than achieving the intended goal (Milvus, 2023). Training instability, especially when fine-tuning large language models, presents another hurdle, with algorithms such as REINFORCE++ (Hu, 2025) often suffering from volatile policy updates. To address these instabilities and enhance training convergence, improved algorithms such as Group-Relative Policy Optimization (GRPO) have been developed, which also help to eliminate reward hacking within RL frameworks for LLMs (Shao et al., 2024). Our proposed AUTOTRITON aligns with this perspective. In the field of kernel generation, we employ the GRPO algorithm and design reasonable rewards to build the LLM's understanding of kernel queries and conduct RL-powered Triton programming accordingly.

## 3 AUTOTRITON

In this section, we introduce AUTOTRITON, a specialized model adapted for the Triton programming task. AUTOTRITON is characterized by a sequential two-stage process. Initially, the model undergoes SFT to establish a strong foundation in Triton programming principles. Following this, an RL

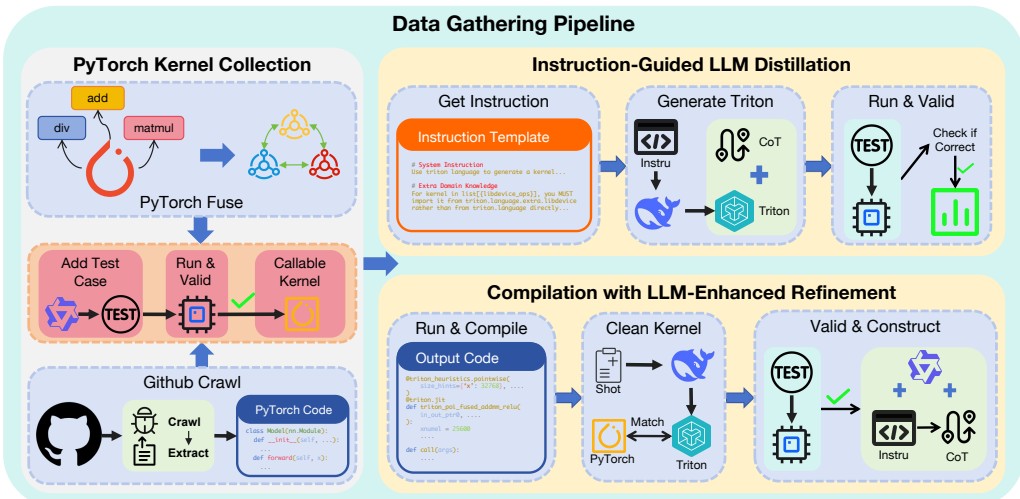

Figure 2: Data gathering pipeline of AUTOTRITON. Our pipeline begins with the **systematic collection of PyTorch kernels**, then generates corresponding Triton kernels by **instruction-guided LLM distillation** and **compilation with LLM enhanced refinement** simultaneously.

framework is applied, which allows execution-based feedback of GPU code, guiding the model to further optimize the generated kernels' correctness and performance. To elucidate AUTOTRITON, we will first formally perform the task formulation, then detail the supervised fine-tuning procedure, and subsequently present the design of the reinforcement learning framework.

### 3.1 PROBLEM FORMULATION

Developing custom kernels traditionally demands substantial domain expertise and involves a significant amount of empirical trial-and-error. To accelerate this development lifecycle, we define the task of Triton programming. This task aims to learn a mapping from a comprehensive kernel specification to its corresponding executable Triton implementation. A kernel specification $\mathcal{D}$, generally comprises two primary components: a concrete PyTorch implementation or a formal interface definition that details its functional description, the signatures of input and output parameters (including data types), and the dimensionality (shapes) of the tensors involved. The core challenge is to develop a model $\mathcal{M}$ that, given $\mathcal{D}$, synthesizes a Triton kernel $\mathcal{T}$ that is not only syntactically correct and executable, but also semantically faithful to all the requirements outlined in $\mathcal{D}$.

### 3.2 SUPERVISED FINE-TUNING

Recent studies (Li et al., 2025) have highlighted that even models proficient in general-purpose programming exhibit limited capabilities to generate specialized Triton kernels. To bridge this capability gap and equip our model with essential Triton programming expertise, we develop a meticulous data gathering pipeline to produce high-quality data for supervised fine-tuning. This pipeline automates the crucial steps of data collection, synthesis, and validation, with the explicit goal of retaining only high-fidelity, syntactically sound, and demonstrably correctly executable data for training. The architecture of the pipeline is illustrated in Figure 2.

Our proposed pipeline for data gathering begins with the **systematic collection of PyTorch kernels**. This entails harvesting kernels from established open-source platforms such as GitHub and Hugging-Face, supplemented by the algorithmic composition of basic kernels through the PyTorch interface. We then leverage an open-source LLM proficient in programming, such as Qwen2.5-Coder (Hui et al., 2024), for the automated generation of test cases, which are subsequently used to validate and retain executable PyTorch kernels.

Following the collection of PyTorch kernels, we employ two distinct strategies to generate their corresponding Triton kernels: **instruction-guided LLM distillation** and **compilation with LLM-enhanced refinement**.

The distillation-based approach involves creating targeted instructions that encapsulate both the PyTorch kernel's functionality and relevant Triton-specific knowledge. A capable deep-reasoning LLM, such as DeepSeek R1 (Guo et al., 2025), is prompted with these instructions to generate Triton code, accompanied by a step-by-step Chain-of-Thought (CoT) explanation. The generated Triton snippets are then cross-validated against the original PyTorch kernels using the previously generated test cases, and only functionally equivalent pairs are selected for supervised fine-tuning.

Recognizing the inherent limitations of general-purpose LLMs in proficiently generating Triton code (Li et al., 2025), we also leverage a compilation-based approach for enhanced data acquisition efficiency. Specifically, PyTorch code snippets are processed using `torch.compile`. The resultant compiled artifacts are then refined by an LLM to improve human readability; this involves tasks such as inserting explanatory comments, removing extraneous decorators, and renaming variables to be more semantically meaningful. After verifying functional equivalence with PyTorch using test cases, we leverage an LLM to craft instructions. These, along with the verified Triton code, are used to prompt an LLM to generate detailed CoT narratives, aiming to instill Triton programming paradigms during supervised fine-tuning.

Finally, the culminating dataset, comprising `<instruction, Triton code with CoT>` pairs, is leveraged for supervised fine-tuning. The model learns to predict the Triton code and its CoT justification conditioned on the instruction prompt. thereby developing foundational capabilities in Triton programming.

### 3.3 REINFORCEMENT LEARNING

To further push the border of the coding ability of AUTOTRITON, we adopt a common Reinforcement Learning with Verifiable Reward (RLVR) pipeline. Our training process is based on the GRPO algorithm (Shao et al., 2024), updating the policy with a group normalized objective:

$$\mathcal{J}_{GRPO}(\theta) = \mathbb{E}[q \sim P(Q), \{o_i\}_{i=1}^{G} \sim \pi_{\theta_{old}}(O|q)]$$

$$\frac{1}{G}\sum_{i=1}^{G}\frac{1}{|o_i|}\sum_{t=1}^{|o_i|}\left\{\min\left[\frac{\pi_\theta(o_{i,t}|q,o_{i,<t})}{\pi_{\theta_{old}}(o_{i,t}|q,o_{i,<t})}\hat{A}_{i,t}, \text{clip}\left(\frac{\pi_\theta(o_{i,t}|q,o_{i,<t})}{\pi_{\theta_{old}}(o_{i,t}|q,o_{i,<t})}, 1-\epsilon, 1+\epsilon\right)\hat{A}_{i,t}\right] - \beta D_{KL}(\pi_\theta||\pi_{ref})\right\}$$

$$(1)$$

where $\pi_\theta$ and $\pi_{\theta_{old}}$ are the policy model and reference model, and $\hat{A}_{i,t}$ is the group-wise advantage:

$$\hat{A}_{i,t} = \frac{r_i - \text{mean}(\{r_j\}_{j=1}^{N})}{\text{std}(\{r_i\}_{i=1}^{N})}.$$

$$(2)$$

The reward function is defined by the following equation, which combines an execution-based component with a rule-based one:

$$R(\hat{a}) = \begin{cases} 1, & \text{is\_Triton}(\hat{a}) \,\&\, \text{test\_passed}(\hat{a}) \\ 0, & \text{otherwise} \end{cases}$$

$$(3)$$

The current training dataset, derived primarily from distillation and compilation, has performance limitations that render it insufficient for performance-guided training. The acquisition of more higher-quality data is reserved for future work.

The data for our RL stage is also generated using the pipeline from § 3.2. In this stage, we only retain `<instruction, PyTorch code>` pairs, as the reference PyTorch code is sufficient for deriving a reward signal through test-case execution, eliminating the need for labeled Triton code. This enables the inclusion of more difficult, out-of-distribution (OOD) data particularly suitable for RL exploration. The final training data is a strategic mix of these novel instances and a small portion of in-distribution data from the SFT phase to ensure a smooth policy transition.

## 4 EXPERIMENTS

In this section, we evaluate the performance of AUTOTRITON and provide a comprehensive analysis from multiple aspects.

## 4.1 EVALUATION SETUP

**Evaluation Benchmarks** We evaluate AUTOTRITON using two established benchmarks: TRITONBENCH (Li et al., 2025) [1] and KERNELBENCH (Ouyang et al., 2025) [2]. TRITONBENCH assesses LLM capabilities in generating Triton kernels, which is divided into two evaluation channels: TRITONBENCH-G consists of $184$ real-world kernels from GitHub and TRITONBENCH-T consists of $166$ kernels aligned with PyTorch interfaces. While KERNELBENCH evaluates LLM proficiency in generating efficient GPU kernels for neural network optimization across $250$ tasks, categorized into Level 1 ($100$ single-kernel tasks, e.g., convolution, for CUDA replacement), Level 2 ($100$ simple fusion tasks, e.g., conv+bias+ReLU, for fused CUDA kernels), and Level 3 ($50$ full architecture tasks, e.g., MobileNet, for end-to-end CUDA optimization). The prompts used during inference are detailed in figure 5.

**Evaluation Metrics** Regarding the evaluation metrics, we synthesize those from the above two benchmarks and categorize them into two aspects: (1) `Compilation Accuracy` (error-free compilations); (2) `Call Accuracy` (error-free invocation); (3) `Execution Accuracy` (correct input-output behavior); (4) `Speed Up` (relative execution time improvement) on both benchmarks. Following the original settings of both benchmarks, we evaluate `Compilation Accuracy`, `Execution Accuracy`, `Speed Up` on KERNELBENCH, and evaluate `Call Accuracy`, `Execution Accuracy`, `Speed Up` on TRITONBENCH respectively. For evaluations on TRITONBENCH-G, the `Speed Up` value is derived by comparing against their supplied reference Triton code, while for all other evaluations, `Speed Up` is calculated against PyTorch implementations. Following KERNELBENCH, we report $\text{fast}_p$ to measure the absolute speedup of Triton codes across the entire benchmark, which is calculated as follows:

$$\text{fast}_p = \frac{1}{N} \sum_{i=1}^{N} \mathbb{1}(\text{correct}_i \wedge \{\text{SpeedUp}_i > p\}), \tag{4}$$

**Training Details** In the SFT stage, we utilize the LLaMA-Factory framework (Zheng et al., 2024) with a dataset of $14,102$ samples. We set the maximum sequence length to $16,384$ and use a training batch size of 1 per device. The model is fine-tuned with a learning rate of $1 \times 10^{-5}$ for 3 epochs. This stage is completed in approximately $16$ hours on a single node with $8$ A800 GPUs. For the subsequent RL stage, we adopt the VeRL framework (Sheng et al., 2025), using the dataset containing $6,302$ samples. In this phase, the training batch size is set to $64$. The maximum prompt length is capped at $4,096$ tokens, while the maximum response length is set to $16,384$ tokens. The learning rate for the actor's optimizer is configured to $1 \times 10^{-6}$. The model is trained for 1 epoch. This phase requires approximately 32 hours of training time on two nodes, utilizing a total of $16$ A800 GPUs.

## 4.2 MAIN RESULTS

Table 1 and Table 2 present the experimental results of AUTOTRITON on TRITONBENCH and KERNELBENCH, respectively. Across five evaluation channels, AUTOTRITON consistently outperforms powerful models such as GPT-4o, Claude-4-Sonnet, Qwen3-32B, DeepSeek-R1-0120, and DeepSeek-V3.1 in both correctness and runtime performance. Furthermore, it achieves competitive performance relative to the most advanced models, including GPT-5 and DeepSeek-R1-0528, highlighting the effectiveness of our proposed data gathering pipeline and training framework in generating high-fidelity training instances.

It is further observed that on the TRITONBENCH-G channel, all evaluated models struggle significantly in both correctness and performance. The inherent difficulty of this channel, which evaluates models on real-world requirements from GitHub against reference Triton implementations, highlights the substantial challenges that persist in automated Triton programming. These findings suggest that the task continues to pose a significant challenge and calls for deeper investigation.

---

[1] We use TRITONBENCH-T version in `https://github.com/thunlp/TritonBench/pull/6`.

[2] We use the Triton backend version of KERNELBENCH in `https://github.com/ScalingIntelligence/KernelBench/pull/35`.

| Model | #Params | TRITONBENCH-G | | TRITONBENCH-T | |
| --- | --- | --- | --- | --- | --- |
| | | Call / Exec | fast$_1$ / fast$_2$ | Call / Exec | fast$_1$ / fast$_2$ |
| Seed-Coder-Reasoning | 8B | 2.72 / 2.72 | 0.54 / 0.00 | 3.61 / 3.61 | 1.20 / 0.00 |
| Qwen3 | 8B | 2.17 / 1.63 | 0.54 / 0.00 | 6.02 / 5.42 | 2.41 / 0.00 |
| Qwen3 | 32B | 10.33 / 9.24 | 2.17 / 1.63 | 21.96 / 21.96 | 10.84 / 3.01 |
| GPT-4o | - | 10.87 / 10.33 | 4.89 / 1.63 | 18.67 / 15.06 | 7.84 / 1.20 |
| GPT-5 | - | 16.30 / 15.76 | 4.89 / 1.09 | 34.34 / 34.34 | 14.46 / 4.22 |
| Claude-4-Sonnet | - | 9.24 / 9.24 | 1.64 / 1.09 | 10.84 / 10.84 | 4.22 / 1.84 |
| DeepSeek-R1-0120 | 671B | 13.59 / 13.05 | 4.89 / 0.54 | 28.92 / 28.37 | **22.89** / 3.01 |
| DeepSeek-R1-0528 | 685B | 16.30 / 15.22 | 3.80 / 0.54 | 30.72 / 30.12 | 11.45 / 4.82 |
| DeepSeek-V3.1 | 671B | **17.93** / **17.39** | 4.89 / 0.54 | 33.73 / 33.73 | 13.25 / 3.61 |
| AUTOTRITON | 8B | 15.76 / 15.76 | **7.61** / **2.17** | **40.36** / **39.16** | **17.04** / 6.02 |
| w/o RL (SFT only) | 8B | 14.67 / 14.13 | 4.89 / 1.09 | 34.94 / 34.94 | 15.06 / **7.83** |

Table 1: Main results on TRITONBENCH. We present `Call Accuracy` (Call), `Execution Accuracy` (Exec), fast$_1$ and fast$_2$. The best-performing and second-best-performing methods are highlighted in **Bold** and Underline, respectively.

| Model | #Params | LEVEL1 | | LEVEL2 | | LEVEL3 | |
| --- | --- | --- | --- | --- | --- | --- | --- |
| | | Comp / Exec | fast$_1$ / fast$_2$ | Comp / Exec | fast$_1$ / fast$_2$ | Comp / Exec | fast$_1$ / fast$_2$ |
| Seed-Coder-Reasoning | 8B | 48.0 / 10.0 | 4.0 / 2.0 | 44.0 / 11.0 | 5.0 / **4.0** | 52.0 / 10.0 | 4.0 / **4.0** |
| Qwen3 | 8B | 52.0 / 16.0 | 9.0 / **9.0** | 73.0 / 16.0 | 8.0 / 1.0 | 40.0 / 14.0 | 4.0 / 2.0 |
| Qwen3 | 32B | 84.0 / 23.0 | 5.0 / 4.0 | 98.0 / 25.0 | 15.0 / 2.0 | **92.0** / 16.0 | 6.0 / 0.0 |
| GPT-4o | - | 91.0 / 15.0 | 3.0 / 1.0 | 83.0 / 5.0 | 3.0 / 0.0 | 74.0 / 8.0 | 4.0 / 2.0 |
| GPT-5 | - | **99.0** / 23.0 | 4.0 / 0.0 | 97.0 / 22.0 | 11.0 / 4.0 | 88.0 / **40.0** | **14.0** / 0.0 |
| Claude-4-Sonnet | - | 87.0 / 33.0 | **11.0** / 7.0 | 92.0 / 26.0 | 10.0 / 1.0 | 82.0 / 18.0 | 2.0 / 0.0 |
| KernelLLM | 8B | 72.0 / 20.2 | − / − | 76.0 / 16.0 | − / − | − / − | − / − |
| DeepSeek-R1-0120 | 671B | 95.0 / 30.0 | 5.0 / 1.0 | 91.0 / 26.0 | 21.0 / 2.0 | 74.0 / 4.0 | 0.0 / 0.0 |
| DeepSeek-R1-0528 | 685B | 90.0 / 35.0 | 7.0 / 1.0 | 90.0 / 42.0 | **28.0** / 2.0 | 76.0 / 26.0 | **14.0** / 2.0 |
| Deepseek-V3.1 | 671B | 97.0 / 26.0 | 7.0 / 4.0 | **99.0** / 26.0 | 16.0 / 0.0 | 86.0 / 18.0 | 8.0 / 2.0 |
| AUTOTRITON | 8B | 83.0 / **36.0** | 10.0 / 6.0 | 97.0 / **45.0** | 17.0 / 0.0 | 82.0 / 20.0 | 10.0 / **4.0** |
| w/o RL (SFT only) | 8B | 65.0 / 29.0 | 10.0 / 4.0 | 85.0 / 27.0 | 8.0 / 3.0 | 64.0 / 6.0 | 2.0 / 2.0 |

Table 2: Main results on KERNELBENCH. We present `Compilation Accuracy` (Comp), `Execution Accuracy` (Exec), fast$_1$ and fast$_2$. The best-performing and second-best-performing methods are highlighted in **Bold** and Underline, respectively.

## 4.3 ANALYSIS

**Cross Comparisons for Triton and CUDA Models**   To further assess the performance of AU-TOTRITON, we provide a comparative analysis against prominent kernel generation models not specifically focused on Triton programming, namely AI CUDA Engineer (Lange et al., 2025) and Kevin-32B (Baronio et al., 2025). We employ the KERNELBENCH benchmark for this evaluation due to its ability to assess both Triton and CUDA kernels, ensuring a fair comparison. As illustrated in Table 3, we report the P75 and P50 speedups over the PyTorch baseline in the pass@10 setting, which represent the speedup ratios at the 75th and 50th percentiles of the kernel performance distribution.

A key finding from our analysis is the persistent, systematic gap in automated programming proficiency between Triton and CUDA, which highlights the formidable challenges associated with high-performance Triton code generation. Even against this backdrop, AUTOTRITON establishes its superiority over the recent specialized AI CUDA Engineer framework, delivering quantitatively better results in metrics of both correctness and runtime efficiency. Although these results validate the strength of AUTOTRITON, it still lags behind the Kevin-32B (Baronio et al., 2025) model, which we attribute to three potential causes: the intrinsic programming model differences between CUDA and Triton, a parameter scale mismatch (8B vs. 32B), and the use of 90% of the evaluation data in Kevin-32B's training, which likely results in higher evaluation scores.

**Effects of Reinforcement Learning**   The final rows of Table 1 and Table 2 present the performance of AUTOTRITON without the RL stage. A clear performance uplift is observed when comparing AUTOTRITON with its SFT-only counterpart, demonstrating that RL effectively raises the performance ceiling for the Triton programming task. This result suggests that RL enables the model to transcend the inherent limitations of imitation learning, consistent with observations in other domains. Furthermore, the performance gains achieved during the RL stage also validate the efficacy of our proposed data gathering pipeline. This pipeline effectively generates a training dataset that is highly

| Model | Lang. | #Params | LEVEL1 Comp / Exec | P75 / P50 | LEVEL2 Comp / Exec | P75 / P50 | LEVEL3 Comp / Exec | P75 / P50 |
|---|---|---|---|---|---|---|---|---|
| AI Cuda Engineer | | | | | | | | |
| - o1-preview | CUDA | - | − / 63.0 | 0.96 / 0.45 | − / **95.0** | 1.01 / 1.00 | − / 19.0 | 1.00 / 0.99 |
| - o1-high | CUDA | - | − / 50.0 | 0.97 / 0.37 | − / 81.0 | 1.00 / 0.87 | − / 12.0 | 1.00 / 0.93 |
| Claude-4-Sonnet | CUDA | - | 99.0 / 64.0 | **1.26** / **0.97** | 100.0 / 92.0 | 1.42 / 1.19 | **100.0** / 66.0 | **1.22** / **1.00** |
| GPT-5 | CUDA | - | 79.0 / 54.0 | 1.13 / 0.81 | 97.0 / 65.0 | 1.46 / 1.16 | 94.0 / 36.0 | 6.0 / 0.0 |
| DeepSeek-R1-0528 | CUDA | 685B | 99.0 / **97.0** | 1.23 / 0.85 | 100.0 / 100.0 | 1.74 / 1.33 | 100.0 / 70.0 | 1.17 / **1.00** |
| DeepSeek-V3.1 | CUDA | - | 99.0 / 83.0 | 1.15 / 0.83 | 100.0 / 90.0 | 1.53 / 1.25 | 98.0 / 50.0 | 1.07 / **1.00** |
| Kevin* | CUDA | 32B | **100.0** / 88.0 | 1.14 / 0.78 | 98.0 / 86.0 | 1.64 / 1.24 | **100.0** / 70.0 | 1.10 / 0.92 |
| KernelLLM | Triton | 8B | 99.0 / 52.0 | − / − | 97.0 / 34.0 | − / − | − / − | − / − |
| Claude-4-Sonnet | Triton | - | 99.0 / 57.0 | 1.01 / 0.76 | 100.0 / 68.0 | 1.41 / 1.12 | 99.0 / 60.0 | 1.12 / **1.00** |
| GPT-5 | Triton | - | **100.0** / 49.0 | 1.14 / 0.83 | 100.0 / 55.0 | 1.32 / 1.08 | 98.0 / 62.0 | 1.02 / 0.90 |
| DeepSeek-R1-0528 | Triton | 685B | **100.0** / 74.0 | 1.04 / 0.62 | 100.0 / 74.0 | 1.56 / 1.28 | **100.0** / **72.0** | 1.03 / 0.92 |
| DeepSeek-V3.1 | Triton | - | **100.0** / 62.0 | 1.13 / 0.90 | 100.0 / 77.0 | 1.40 / 1.16 | 98.0 / 64.0 | 1.08 / 1.00 |
| AUTOTRITON | Triton | 8B | **100.0** / 68.0 | 1.01 / 0.69 | **100.0** / 88.0 | 1.17 / 1.01 | 88.0 / 52.0 | 1.03 / **1.00** |

Table 3: Cross comparison results for Triton and CUDA models on KERNELBENCH. We report pass@10 results for each model. We present `Compilation Accuracy` (Comp), `Execution Accuracy` (Exec), `Torch P75` (P75) and `Torch P50` (P50). The best-performing and second-best-performing methods are highlighted in **Bold** and Underline, respectively. * denotes that they use 180 of the evaluation data for training purpose.

| Model | TRITONBENCH-T | KERNELBENCH-Level 1 |
|---|---|---|
| AUTOTRITON | 5 | 6 |
| w/o rule-based reward | 18 | 25 |
| w/o RL (SFT only) | 4 | 4 |
| w/o SFT&RL (Backbone model) | 66 | 10 |

Table 4: Numbers of generated Triton code that do not contains keyword "`@triton.jit`".

suitable for exploration during RL, which plays a crucial role in pushing the boundaries of the Triton programming task.

**Effects of Reward Design**    As mentioned in § 3.3, a primary challenge in the Triton programming task is reward hacking, where models learn to satisfy test cases without generating correct Triton code. To address this, we introduce auxiliary rule-based rewards alongside the primary execution-based reward to explicitly incentivize adherence to the Triton language specification. The impact of this strategy is quantified in Table 4. By checking for the mandatory "`@triton.jit`" decorator, we find that rule-based rewards significantly decrease the count of invalid generations on TRITONBENCH-T (from 18 to 5) and KERNELBENCH-Level1 (from 25 to 6), confirming the importance of explicit syntactic guidance in the reward mechanism. Despite these improvements, a rule-driven reward function can still be hacked. Models may learn to generate low-quality code that satisfies the explicit rules but fails to fulfill the complete semantic requirement of Triton. For example, as illustrated in Figure 3, when tasked with implementing a kernel composed of a convolution and a ReLU (Figure 3(a)), the model often generates a valid Triton kernel for the simpler ReLU part while leaving the more complex convolution as a fallback PyTorch implementation (Figure 3)(b). More critically, the model might circumvent the reward rules entirely by generating a fake Triton kernel that it never calls (Figure 3(c)). Low-quality implementations are highly prevalent across all evaluated models. A potential countermeasure involves incorporating runtime-based performance rewards, which we reserve for future work.

**Effects of Supervised Fine-tuning**    As shown in Table 1 and Table 2, after undergoing SFT, the model achieves superior performance compared to the original backbone model (Seed-Coder-8B-Reasoning). This initial result suggests that SFT is effective in familiarizing the model with the fundamental paradigms of Triton programming and further proves the effectiveness of our proposed data gathering pipeline in generating high-quality SFT data. This conclusion is further supported by the training dynamics in Figure 4. Although the model without SFT also shows a notable upward trend in its reward curve, it suffers from severe reward hacking, and the majority of instances show the behavior illustrated in Figure 3(c). Specifically, although the generated code passes test cases, it often deviates from Triton's syntax, defaulting to simpler PyTorch implementations, consistent with the trend in Table 4. This highlights that SFT is essential not only for learning correct syntax, but also

```
Original PyTorch Kernel

class Model(nn.Module):
    """
    Simple model that performs a convolution,
applies ReLU, and adds a bias term.
    """

    def __init__(self, in_channels, out_channels,
            kernel_size, bias_shape):
        super(Model, self).__init__()
        self.conv = nn.Conv2d(in_channels,
            out_channels, kernel_size)
        self.bias = nn.Parameter(torch.randn(
            bias_shape))

    def forward(self, x):
        x = self.conv(x)
        x = torch.relu(x)
        x = x + self.bias
        return x
```
(a)

```
Triton Kernel

@triton.jit
def triton_relu_add_kernel(in_out_ptr, bias_ptr,
        num_elements, XBLOCK: tl.constexpr):
    pid = tl.program_id(axis=0)....

def triton_relu_add_bias(x, bias):
    ....
    grid = (triton.cdiv(x_flat.numel(), 1024),)
    triton_relu_add_kernel[grid](x_flat,
        self.bias.view(-1), x_flat.numel(), 1024)
    ....

class ModelNew(nn.Module):
    def __init__(self, in_channels, out_channels,
            kernel_size, bias_shape):
        super(Model, self).__init__()
        self.conv = nn.Conv2d(in_channels,
            out_channels, kernel_size)
        self.bias = nn.Parameter(torch.randn(
            bias_shape))

    def forward(self, x):
        x = self.conv(x)
        x = triton_relu_add_bias(x)
        return x
```
(b)

```
Triton Hacking Kernel

@triton.jit
def triton_relu_add_bias(in_out_ptr, bias_ptr,
        num_elements, XBLOCK: tl.constexpr):
    pass

class ModelNew(nn.Module):
    def __init__(self, in_channels, out_channels,
            kernel_size, bias_shape):
        super(Model, self).__init__()
        self.conv = nn.Conv2d(in_channels,
            out_channels, kernel_size)
        self.bias = nn.Parameter(torch.randn(
            bias_shape))

    def forward(self, x):
        x = self.conv(x)
        x = torch.relu(x)
        x = x + self.bias
        return x
```
(c)

Figure 3: Example of the phenomenon of the low-quality implementation of Triton code.

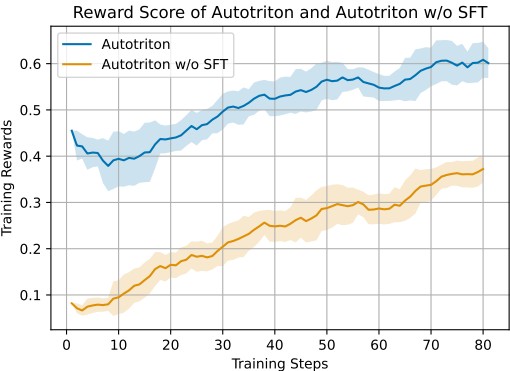

Figure 4: Reward scores of AUTOTRITON and AUTOTRITON w/o SFT stage.

for preventing reward hacking, where the model exploits test cases with trivial Torch code instead of mastering genuine Triton programming.

**Limitations**  A key limitation of AUTOTRITON is that the current training framework does not incorporate performance-guided training. This is because, at the current stage, the available training data is not sufficient to reliably support performance-guided training. In future work, we plan to collect more higher-quality data to enable performance-guided training, allowing AUTOTRITON to generate more efficient kernels optimized for the target hardware.

## 5 CONCLUSION

In this work, we propose AUTOTRITON, the first RL-powered model dedicated to Triton programming. AUTOTRITON involves a two-stage training process: an SFT stage where AUTOTRITON learns essential Triton programming expertise from high-quality data generated by our novel data curation pipeline, followed by an RL stage where it further improves by exploring more challenging problem instances. Evaluations in five channels of TRITONBENCH and KERNELBENCH show that AUTOTRITON achieves performance comparable to mainstream LLMs, including GPT-5 and DeepSeek-R1-0528. Our in-depth analysis of each component validates the significant potential of RL-based methods for automatic Triton programming. Ultimately, AUTOTRITON demonstrates a promising pathway toward the automated generation of efficient kernels, offering a new paradigm for building high-performance AI systems.

ETHICS STATEMENT

This work focuses on automated Triton kernel generation using reinforcement learning. It does not involve human subjects, private data, or sensitive attributes. The data used in this study are publicly available or synthetically generated, and we follow standard research integrity practices.

The potential benefits of our work include lowering the barrier to high-performance GPU programming and improving the efficiency of AI workloads. At the same time, we acknowledge risks: automatically generated kernels may contain correctness or efficiency issues, and automated system-level programming could be misused for unintended purposes. We encourage responsible adoption of our approach, including careful verification and testing before deployment.

REPRODUCIBILITY STATEMENT

The methodology, training settings, and evaluation protocols are described in detail in the Methodology and Experiments sections. Following these instructions, researchers can reproduce our results and train models with comparable performance. Additional resources will be released after publication.

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

## A  INFERENCE PROMPTS

Figure 5: AUTOTRITON prompts for experimental reasoning.

## B  LLM Usage

We used LLM solely for language refinement, such as grammar checking and minor wording improvements. The model did not contribute to research ideation, experimental design, implementation, analysis, or writing of the scientific content.

