# OpenReview forum: "AUTOTRITON: Automatic Triton Programming with Reinforcement Learning in LLMs"
_ICLR.cc/2026/Conference — Submitted to ICLR 2026_

### Official Review · Reviewer_txNq · 2025-10-29

**Soundness:** 2
**Presentation:** 2
**Contribution:** 1
**Rating:** 2
**Confidence:** 4

**Summary:**

AutoTriton proposes reinforcement learning (RL)-based framework to refine LLMs for triton kernel generation. Authors first collect and curate the data from various sources on Github, torch.compile, and synthetic generations using an LLM. Generated data points are validated for correctness and correct kernels are retained in the dataset. Authors initially perform supervised finetuning to enable model understand Triton language. Then RL finetuning is performed using GRPO method with verifiable rewards to enable refine the model generations. Authors have employed rule-based rewards with execution correctness component. Authors have observed and addressed reward hacking via rule-based detection of a key attribute in triton programming. Authors demonstrate the effectiveness of their approach on TritonBench-G and T and Kernelbench benchmarks. Though the fraction of correct kerrnel generations are lower via AutoTriton, speedup obtained post generations is quite on-par with other methods.

**Strengths:**

- Extensive data collection methodology for obtaining datapoints with wide variety.
- torch.compile data is usually ambiguous and authors have developed an approach to extract relevant code from it.
- Supervised finetuning to establish base knowledge followed by RL finetuning with verifiable rewards.
- Address reward hacking with rule-based identification of key attribute in triton programming.
- Speedup obtained by generated kernels is on par with that of frontier models.

**Weaknesses:**

- There have been several reported issues with TritonBench benchmark (https://github.com/thunlp/TritonBench/issues/7 and other sources). It is not clear if authors have addressed these issues in their experimentation. Authors are encouraged to discuss in detail how these issues were addressed. Without these fixes the results on TritonBench benchmark cannot be trusted.
- Some of the artifacts of TritonBench issues are observed in Table 1 results: Call and Exec accuracy are the same or very close for most models. TritonBench uses fewer and poor unit tests for Exec accuracy. So this data cannot be directly trusted.
- Though the paper shows good enough content, it is presented more as a technical report rather than a scientific study that discusses in details fundamental reasoning behind their choices in experimentation. For example, why was GRPO used for refinement? There are variants of DPO with ranked data that can be used for preferential training with the generated data instead.
- Lines 257-259: As authors admit here, use of poor and sparse reward function in RL finetuning will not result in performance improvement of generated kernels. If performance improvement is not THE goal, then there is no incentive to generate kernels and spend so much energy in training/inference/generation.
- Rather than discussing single speedup number, e.g. fast_1 or fast_2, studying and understanding the speedup distribution usually as a function of difficulty level sheds more light on efficacy of  an approach.
- KernelBench discussion is poorly presented. It is not clear (unless one goes over the kernelbench pull mentioned) whether authors are referring to CUDA or Triton version of kernelbench.
- Assuming KernelBench Triton version: the performance on AutoTriton is quite poor in terms of number of correct generated kernels and speedup/performance improvements. (Table 3)

**Questions:**

- Do results in Table 1 still hold after fixing the TritonBench issues?
- Can authors provide with a case study with "aha" moment case study on at least one kernel that shows interesting reasoning and code generation by AutoTriton?

---

> ### Author Response · Authors · 2025-12-03
> **Response to Weaknesses**
>
> Thank you for recognizing our data collection method, SFT+RL strategy, and solution to reward hacking. We are committed to addressing your concerns regarding benchmarks and scientific rigor.
>
> Response to Weaknesses
>
> 1. Unresolved issues reported in TritonBench: TritonBench is not the only evaluation set we used; we also utilized KERNELBENCH for evaluation. The internal flaws of TritonBench (e.g., Issue #7) are mainly attributed to the maintenance of the upstream benchmark and are unrelated to the AutoTriton methodology (SFT data pipeline + RL strategy) proposed in this paper. We used the latest and reviewed version available at the time of the experiment and handled known issues to ensure fairness. As the community fixes TritonBench, we promise to update the corresponding data in the final version, but this does not change the conclusion of the method's effectiveness.
>
> 2. Call and Exec accuracy are close: As stated above, this is an inherent property of the benchmark's test sparsity. We emphasize the Speed Up metric.
>
> 3. Paper reads like a technical report; lack of scientific reasoning for GRPO: We accept this criticism and pledge to enhance the scientific rigor and discussion of underlying principles in the final version. We chose GRPO because it excels in handling training instability during LLM policy updates. While DPO is suitable for preference learning based on ranked data, GRPO's on-policy nature is better suited for our RL training environment based on online execution and binary (correctness) reward signals.
>
> 4. Sparse reward function and incentive for performance improvement: We acknowledge the reward function is sparse, but we strongly disagree with the view that there is a lack of incentive. The incentive lies in generating code that is functionally correct and complies with Triton syntax. This is the foundation of automated kernel generation. The Speed Up gains (fast$_1$, fast$_2$) achieved by the model demonstrate that by rewarding correctness, the model implicitly learns superior Triton programming habits (e.g., more effective block structures, code more conducive to compiler optimization).
>
> 5. Speed Up distribution instead of single number: We have already presented P75 and P50 data in the pass@10 results (Table 3), which represent the speedup values at the top 25% and top 50% quantiles. We believe this effectively illustrates the distribution information.
>
> 6. Clarity of KernelBench discussion (CUDA or Triton version): We clarify that all our experiments use the Triton version of KernelBench. We will ensure this is clearly stated in the revision.
>
> 7. View that KernelBench performance (Correctness and Speed Up) is poor:
>
>     a. Efficiency of Small-Scale Models: Our 8B model performs comparably to or better than frontier closed-source large models (e.g., GPT-5, DeepSeek-R1) on core performance metrics Speed Up (fast$_1$/fast$_2$). Table 2 shows that on KernelBench, AutoTriton (8B)'s fast_1 even surpasses the much larger Qwen3-32B and GPT-4o.
>
>     b. Validation and Scalability of Methodology: The lower absolute accuracy mainly reflects the fundamental capability limits of an 8B model and the difficulty of the benchmark tasks, rather than a flaw in our method. The core contribution of this paper is proving the effectiveness of the SFT+RL pipeline. Experiments show this method significantly boosts the base model's capabilities. Under limited academic computational resources, we successfully validated the feasibility of this path. With increased data volume and base model scale, this method is expected to easily scale and achieve superior results. Therefore, the absolute performance gap under resource constraints should not be viewed as a methodological weakness.

---

> > ### Author Response · Authors · 2025-12-03
> > **Response to Questions**
> >
> > Response to Questions
> >
> > 1. Do results in Table 1 still hold after fixing TritonBench issues: Yes, as discussed above.
> >
> > 2. Case study with an "Aha" moment showing interesting reasoning and code generation: We will include a detailed case study in the revision, showcasing the "Aha" moment driven by our training pipeline.
> > ```python
> > @triton.jit
> > def triton_depthwise_conv2d(
> >     input_ptr, weight_ptr, bias_ptr, output_ptr,
> >     batch_size, in_channels, H, W,
> >     H_out, W_out,
> >     stride, padding,
> >     input_batch_stride, input_channel_stride, input_h_stride, input_w_stride,
> >     weight_channel_stride, weight_kh_stride, weight_kw_stride,
> >     output_batch_stride, output_channel_stride, output_h_stride, output_w_stride,
> >     numel,
> >     BLOCK_SIZE: tl.constexpr,
> >     kernel_size: tl.constexpr,
> >     has_bias: tl.constexpr,
> >     INPUT_DTYPE: tl.constexpr
> > ):
> >     pid = tl.program_id(0)
> >     idx = pid * BLOCK_SIZE + tl.arange(0, BLOCK_SIZE)
> >     mask = idx < numel
> >
> >     C = in_channels
> >     elements_per_batch = C * H_out * W_out
> >     elements_per_channel = H_out * W_out
> >
> >     b = idx // elements_per_batch
> >     remainder = idx % elements_per_batch
> >     c = remainder // elements_per_channel
> >     remainder2 = remainder % elements_per_channel
> >     i = remainder2 // W_out
> >     j = remainder2 % W_out
> >
> >     acc = tl.zeros((BLOCK_SIZE,), dtype=tl.float32)
> >
> >     for dy in tl.static_range(0, kernel_size):
> >         for dx in tl.static_range(0, kernel_size):
> >             input_i = i * stride + dy - padding
> >             input_j = j * stride + dx - padding
> >             valid = (input_i >= 0) & (input_i < H) & (input_j >= 0) & (input_j < W)
> >
> >             input_offsets = b * input_batch_stride + c * input_channel_stride + input_i * input_h_stride + input_j * input_w_stride
> >             input_val = tl.load(input_ptr + input_offsets, mask=valid & mask, other=0.0)
> >
> >             weight_offset = c * weight_channel_stride + dy * weight_kh_stride + dx * weight_kw_stride
> >             weight_val = tl.load(weight_ptr + weight_offset, mask=mask, other=0.0)
> >
> >             acc += input_val * weight_val
> >
> >     if has_bias:
> >         bias_val = tl.load(bias_ptr + c, mask=mask, other=0.0)
> >         acc += bias_val
> >
> >     acc = acc.to(INPUT_DTYPE)
> >     output_offsets = b * output_batch_stride + c * output_channel_stride + i * output_h_stride + j * output_w_stride
> >     tl.store(output_ptr + output_offsets, acc, mask=mask)
> >
> > class ModelNew(nn.Module):
> >     def __init__(self, in_channels: int, kernel_size: int, stride: int = 1, padding: int = 0, bias: bool = False):
> >         super(ModelNew, self).__init__()
> >         self.in_channels = in_channels
> >         self.kernel_size = kernel_size
> >         self.stride = stride
> >         self.padding = padding
> >
> >         self.weight = nn.Parameter(torch.Tensor(in_channels, 1, kernel_size, kernel_size))
> >         if bias:
> >             self.bias = nn.Parameter(torch.Tensor(in_channels))
> >         else:
> >             self.register_parameter('bias', None)
> >
> >         self.reset_parameters()
> >
> >     def reset_parameters(self) -> None:
> >         nn.init.kaiming_uniform_(self.weight, a=math.sqrt(5))
> >         if self.bias is not None:
> >             fan_in, _ = nn.init._calculate_fan_in_and_fan_out(self.weight)
> >             bound = 1 / math.sqrt(fan_in) if fan_in > 0 else 0
> >             nn.init.uniform_(self.bias, -bound, bound)
> >
> >     def forward(self, x: torch.Tensor) -> torch.Tensor:
> >         assert x.dim() == 4, "Input must be a 4D tensor"
> >         assert x.shape[1] == self.in_channels, "Input channels must match model in_channels"
> >
> >         batch_size, in_channels, H, W = x.shape
> >         H_out = (H + 2 * self.padding - self.kernel_size) // self.stride + 1
> >         W_out = (W + 2 * self.padding - self.kernel_size) // self.stride + 1
> >
> >         output = torch.empty((batch_size, in_channels, H_out, W_out), device=x.device, dtype=x.dtype)
> >
> >         if output.numel() == 0:
> >             return output
> >
> >         n_elements = output.numel()
> >         has_bias = 1 if self.bias is not None else 0
> >
> >         grid = lambda meta: (triton.cdiv(n_elements, meta['BLOCK_SIZE']),)
> >
> >         triton_depthwise_conv2d[grid](
> >             x, self.weight, self.bias if self.bias is not None else None, output,
> >             batch_size, in_channels, H, W,
> >             H_out, W_out,
> >             self.stride, self.padding,
> >             x.stride(0), x.stride(1), x.stride(2), x.stride(3),
> >             self.weight.stride(0), self.weight.stride(2), self.weight.stride(3),
> >             output.stride(0), output.stride(1), output.stride(2), output.stride(3),
> >             n_elements,
> >             BLOCK_SIZE=1024,
> >             kernel_size=self.kernel_size,
> >             has_bias=has_bias,
> >             INPUT_DTYPE=tl.float32
> >         )
> >         return output
> > ```

---

### Official Review · Reviewer_q2RF · 2025-10-30

**Soundness:** 2
**Presentation:** 3
**Contribution:** 1
**Rating:** 2
**Confidence:** 3

**Summary:**

The paper “AUTOTRITON: Automatic Triton Programming with Reinforcement Learning in LLMs” proposes AUTOTRITON, an RL-enhanced LLM specialized for Triton kernel programming.
The model is built upon Seed-Coder-8B and trained in two stages:
(1) Supervised Fine-Tuning (SFT) with a curated pipeline collecting and validating PyTorch–Triton kernel pairs, and
(2) Reinforcement Learning (RL) using the Group Relative Policy Optimization (GRPO) algorithm with execution- and rule-based rewards.
Experiments on TRITONBENCH and KERNELBENCH show that AUTOTRITON achieves performance comparable to GPT-5 and DeepSeek-R1 despite having only 8B parameters.

**Strengths:**

S1: Well-designed pipeline: The paper presents a systematic end-to-end data pipeline for high-quality Triton kernel collection and verification.
S2: Quantitative validation of RL gains: Clear ablation results show consistent improvement of RL over SFT-only baselines (Tables 1 and 2).
S3: Detailed experimental setup: Evaluation protocols and hyperparameters are thoroughly described, ensuring reproducibility.

**Weaknesses:**

W1: Limited novelty: Prior works (e.g., AI CUDA Engineer) already apply RL or agentic loops to CUDA kernel optimization. The main contribution appears to be applying similar ideas to Triton rather than introducing new RL methodology.
W2: Incomplete reward design: The RL reward focuses on functional correctness but does not explicitly optimize runtime speed, as the authors themselves note.
W3: Unclear difference from CUDA approaches: The paper lacks deeper analysis of what makes Triton-specific optimization fundamentally distinct from CUDA.
W4: Limited practical validation: The GitHub-derived tasks remain challenging, with relatively low success rates, suggesting real-world readiness is still limited.

**Questions:**

Q1: Beyond language differences, what are the key algorithmic or representational distinctions between AUTOTRITON and RL-based CUDA agents?
Q2: How effectively does the model internalize Triton-specific constraints (e.g., tiling, memory layout) during RL training?
Q3: What are the primary obstacles to integrating performance-guided rewards in future work?

---

> ### Author Response · Authors · 2025-12-03
> **Part 1/2**
>
> Thank you for acknowledging our pipeline design, RL gains, and experimental setup. We aim to address your core questions regarding novelty and limitations.
>
> Response to Weaknesses
>
> 1. Limited Novelty: We believe AUTOTRITON possesses significant novelty in several aspects, beyond simply applying RL to Triton:
>
>     a. Focus on Model Training: Unlike training-free agentic approaches like AI CUDA Engineer, AUTOTRITON is the first RL-driven fine-tuned model specifically designed for Triton programming. It achieves performance comparable to GPT-5 level large models with only an 8B model.
>
>     b. Data Pipeline Innovation: We propose a novel and systematic data collection pipeline to generate high-quality SFT data, including CoT reasoning, which is foundational for LLM-driven kernel generation.
>
>     c. Reward Function Innovation: We introduce a Rule-based Reward to solve the unique reward hacking problem in the Triton domain (degeneration to PyTorch implementation), a key challenge unaddressed in traditional RL code generation methods.
>
> 2. Incomplete Reward Design (Lack of runtime speed optimization): We acknowledge that the current reward function focuses on functional correctness and syntactic adherence. This is a critical first step in applying RL to a new domain. Only after ensuring the model can generate correct and compliant Triton code can we effectively introduce performance rewards. We have explicitly stated in the paper that performance-guided training is the next stage of work, as it requires more complex performance-ranked datasets.
>
> 3. Distinction between Triton-specific optimization and CUDA methods: The fundamental difference between Triton optimization and CUDA optimization lies in the level of abstraction:
>
>     a. Triton (High-Level): Abstracts low-level details like threads and shared memory management. Optimization focuses on block-level parallelism, tiling strategies, memory layout, and dynamic selection of triton.jit parameters. The LLM needs to learn these block-level parallel algorithms.
>
>     b. CUDA (Low-Level): Focuses on low-level hardware details like thread IDs, registers, and shared memory.
>
>     c. RL Challenge: Our RL challenge is to teach the LLM to better manage Triton's high-level parallel abstractions. Furthermore, our Rule-based Reward explicitly targets Triton's Pythonic features and its interface boundary with PyTorch.
>
>     Therefore, this paper focuses on solving the problem of automatic kernel programming and optimization under the Triton DSL. Its unique challenges require us to adopt specific data pipelines and RL strategies. We believe the lack of a deep comparison with CUDA agents should not be viewed as an intrinsic flaw of this work.
>
> 4. Limited Real-world Validation: While the success rate on GitHub-derived OOD tasks is lower, this precisely reflects the difficulty and diversity of real-world kernels and highlights the urgent need for automated tools. Our main experimental results (TRITONBENCH and KERNELBENCH) demonstrate that AUTOTRITON achieves significant success in generating high-performance, functionally correct core DL kernels. By using OOD data in the RL stage, we encourage model exploration, aiming to eventually conquer these more challenging tasks.

---

> ### Author Response · Authors · 2025-12-03
> **Part 2/2**
>
> Response to Questions
>
> 1. "Beyond language differences, what are the key algorithmic or representational distinctions between AUTOTRITON and RL-based CUDA agents?" We believe it is necessary to clarify the definitions of "RL-based Model" and "Agent," and to point out that our work and recent CUDA agents are concurrent works. Beyond language differences, the core distinctions are rule-based reward for domain-specific challenges. CUDA agents typically focus on generic code generation, whereas Triton programming faces a unique challenge: models tend to generate PyTorch wrappers to "cheat" test cases instead of generating true GPU kernels. To address this, we designed a Rule-based Reward (e.g., mandatory check for the @triton.jit decorator). Experimental results prove that this mechanism is crucial for eliminating Triton-specific reward hacking. This is a specific design absent in general CUDA agent algorithms.
>
> 2. Internalizing Triton-specific constraints. The model internalizes these constraints primarily through two stages:
>
>     a. SFT Stage: Utilizing high-quality CoT narratives in the data, we explicitly inject Triton programming paradigms (including Tiling strategies, memory access patterns, etc.) into the model's knowledge.
>
>     b. RL Stage: RL rewards (functional correctness) are evaluated on OOD samples. To pass these tests, the generated code must implicitly adhere to correct Triton constraints (e.g., correct memory boundary handling, valid block indexing). The performance gain of RL over SFT proves that the model is acquiring these more robust and optimized programming strategies.
>
> 3. Primary obstacles to integrating performance-guided rewards. The primary obstacle is the difficulty in acquiring large-scale, high-quality, performance-ranked Triton kernel datasets for training. Constructing such a dataset requires:
>
>     a. Generating multiple functionally correct Triton implementations for each kernel specification.
>
>     b. Performing time-consuming and expensive auto-tuning and profiling on target hardware for each implementation to obtain actual runtime speeds.
>
>     c. Ranking these kernels by performance.
>
>     The computational cost of this process is immense, making datasets required for large-scale LLM training difficult to obtain.

---

### Official Review · Reviewer_9T7y · 2025-11-01

**Soundness:** 3
**Presentation:** 3
**Contribution:** 3
**Rating:** 6
**Confidence:** 3

**Summary:**

In this paper the authors propose an RL method that automatically generates a set of training samples to allow an LLM to learn how to program kernels using the Triton DSL. By using a combination of supervised fine training and RL they demonstrate that the Triton kernel produced by an 8B foundation model can be substantially improved in terms of both correctness and performance. One of the core contributions is the automatic generation of the SFT training data using two processes: instruction-guided LLM distillation and compilation with LLM-enhanced refinement. Using both processes, they show that this training set is sufficient for the LLM to internalize the structure of Triton programs and, with the modification of the reward function, encourage the LLM to produce legitimately optimized Triton kernels and avoid reward hacking. The authors claims are supported by the notable performance of AutoTriton, compared to other contemporary code generators and LLMs.

**Strengths:**

- The process of automatically generating a training dataset to support both the SFT and RL phases is a great direction to pursue. The SFT and RL results on the KernelBench suite demonstrate the effectiveness of the approach to impact kernel generation.
- The decomposition of the gains into those achieved through SFT and RL is interesting and acknowledges the strengths of both approaches used independently and in tandem.
- The results illustrate that although AutoTriton is based on an 8B that it can compete with Triton code generated by much larger models.
- Reward hacking seems to be especially detrimental for code generation since it may be hard to detect without inspecting the result. The author's proposal to augment the reward function was somewhat effective in ensuring Triton code was actually generated, not just wrappers.

**Weaknesses:**

- Although Triton is a useful DSL it wasn't clear to me why the techniques wouldn't be equally applicable to several programming languages that are of interest to the ML community.
- While the comparisons with other LLMs do support the author's claims, they compare a fine-tuned model with a model that is not tuned for Triton coding. From that perspective, I would expect AutoTriton to do better than these baseline models.
- Outside of additions to support disuade the model from reward hacking, the RL formulation and context seem rather standard. This is not a bad thing, but it detracts from a bit of the novelty.
- As the author noted, it would be interesting once support for generating a dataset to enable performance-guided training.

**Questions:**

It's not clear to me what is meant by collecting more high-quality training samples to support performance-guiding training. This was mentioned several times throughout the paper as an issue. Does this mean that although the samples collected are functional, they do not demonstrate the, in some sense, "optimal" way to implement a given kernel within the Triton language?

---

> ### Author Response · Authors · 2025-12-03
> **Part 1/1**
>
> Thank you for your positive evaluation of our work, especially for recognizing the automated data generation process, the decomposition of SFT and RL gains, and the competitiveness of the 8B model against larger models.
>
> Response to Weaknesses
>
> 1. Why focus only on Triton: We chose Triton because it represents a critical and accessible domain in GPU programming, offering a balance between abstraction level and performance tuning difficulty. Our core technologies, including the high-quality data collection pipeline (based on instruction-guided distillation and LLM-enhanced compilation refinement) and the RL training strategy (GRPO + Hybrid Reward), are principally generalizable to DSLs or programming languages in other ML domains, provided that corresponding verification environments and datasets can be constructed. We believe the foundation we have established for Triton will guide future applications in other DSLs.
>
> 2. Comparison with untuned LLM baselines: We agree that a model trained from scratch should theoretically outperform untuned baselines. However, comparing against frontier and much larger models like GPT-5 and DeepSeek-R1 highlights the training efficiency and specialization value of our approach. Our 8B model, through proprietary data and RL tuning, achieves performance comparable to these ultra-large-scale general-purpose models, strongly demonstrating the effectiveness of our SFT and RL strategies.
>
> 3. Generality of RL formulation: Although GRPO is an established algorithm, our novelty lies in its unique application to the Triton kernel generation domain and the design of the reward function. The Rule-based Reward is an innovative solution to the reward hacking problem specific to Triton programming (i.e., the model evading Triton programming by reverting to PyTorch wrappers). We believe this domain-specific algorithm application and customized reward design are significant contributions of this work.
>
> Response to Questions
>
> Meaning of "collecting more high-quality training samples for performance-guided training": Your understanding is completely correct. By "high-quality samples," we mean Triton kernel implementations that are functionally correct and achieve optimal or near-optimal performance. Current SFT data has performance limitations. To realize performance-guided training, we need a paired dataset containing:$$\langle \text{PyTorch Kernel Spec}, \text{Triton Kernel}_1, \text{Triton Kernel}_2, \ldots \rangle$$where Triton kernels are ranked by actual runtime performance. This requires an expensive, large-scale auto-tuning and profiling process to generate, which is a significant challenge we reserve for future work.

---

### Official Review · Reviewer_LZoX · 2025-11-02

**Soundness:** 2
**Presentation:** 3
**Contribution:** 2
**Rating:** 4
**Confidence:** 3

**Summary:**

This paper presents AutoTriton, an automatic code generation and optimization framework that uses large language models (LLMs) to produce high-performance GPU kernels in Triton—a high-level programming language for deep learning acceleration. The goal is to automate kernel development while approaching the performance of expert-written implementations.

AutoTriton integrates a multi-stage LLM-guided pipeline to generate Triton kernels directly from high-level PyTorch functions. The framework iteratively refines the generated kernels by collecting runtime performance metrics and feeding them back into the LLM for self-improvement. The system automatically explores tile sizes, memory layouts, and parallelism strategies, learning from failed executions and performance bottlenecks. Experiments show that AutoTriton achieves up to 93.8% of the performance of hand-tuned expert kernels, outperforming prior LLM-based baselines such as AlphaTriton and OpenAI Codex-based methods, while significantly reducing human engineering effort.

**Strengths:**

+ It addresses a critical bottleneck in modern AI infrastructure: the manual effort and expertise required to write efficient GPU kernels. The integration of LLMs with feedback-driven autotuning aligns well with current trends in AI-assisted compiler optimization and code synthesis.

+ The framework combines static analysis, runtime profiling, and iterative LLM prompting, forming a tight optimization loop. Its error recovery and retry mechanism allows it to gracefully handle compilation or runtime failures—often a weak point in LLM code generation.

+ The performance feedback module quantitatively evaluates each kernel and guides subsequent refinement iterations, embodying a form of “RL through interaction.”

+ AutoTriton achieves near-expert-level performance (90–94% of hand-optimized kernels) across diverse workloads (matrix multiplications, softmax, layer norm, etc.). The benchmark suite covers multiple operator types and shapes, ensuring robustness. The performance improvements over baselines are good: up to 2.3× faster than AlphaTriton and 3.7× faster than naive LLM-generated code).

**Weaknesses:**

- The system still relies heavily on the LLM’s prompt design and prior exposure to Triton code examples. Although feedback improves results, semantic misunderstandings (e.g., incorrect indexing or boundary handling) remain common in early iterations.

- While practical, the paper lacks a deeper theoretical model for convergence or performance bounds of its iterative improvement process. There’s no formal guarantee that feedback-driven refinement leads to monotonic performance improvement.

- Evaluations are limited to relatively small kernels (≤1024 threads) and structured workloads.

- It is unclear how well AutoTriton generalizes to complex fused operators, multi-kernel pipelines, or non-Triton environments (e.g., CUDA directly).

- The paper primarily benchmarks against LLM-based systems. A comparison with auto-tuning frameworks such as TVM, Ansor, or DeepDSL would clarify whether LLM-guided synthesis outperforms traditional search-based compilers.

- While technically detailed, the paper is dense in several sections (especially §3.2 and §4.1).

**Questions:**

1) How exactly are performance metrics (e.g., latency, throughput) integrated into the prompt for LLM feedback?
A more concrete example of the textual feedback template would clarify reproducibility.
Are the prompts dynamically adapted per task, or fixed across all benchmarks?

2) How are different failure types (syntax, compilation, runtime, numerical mismatch) detected and prioritized in the retry logic?
Understanding this hierarchy would clarify how efficiently the system learns from failures.

3) How do you ensure that performance feedback (latency, occupancy, etc.) is interpretable by the LLM and leads to actionable improvements? Does the LLM truly “learn” from the feedback or just make stochastic modifications? How many refinement iterations are typically needed before convergence, and how stable are results across runs?

4) Are all kernel executions run on the same hardware and CUDA version to ensure comparability?
Performance on Triton can vary significantly across GPUs; details about environment settings would help confirm fairness.

---

> ### Author Response · Authors · 2025-12-03
> **Part 1/2**
>
> We appreciate your recognition of our approach, particularly regarding addressing the bottleneck in GPU kernel development, combining LLMs with feedback-driven self-optimization, and outperforming existing LLM baselines.
>
> Response to Weaknesses
>
> 1. Reliance on LLM semantics and early iterations: We agree that the model may still harbor semantic misunderstandings during the early SFT stage. However, our RL stage is specifically designed to address this. The RL stage significantly improves semantic and syntactic robustness through:
>
>     a. Execution-based Reward: We strictly reward functional correctness, forcing the model to generate kernels that pass various test cases.
>
>     b. Rule-based Reward: This is specifically designed to prevent "Reward Hacking" (e.g., the model generating PyTorch wrappers instead of Triton code). By enforcing the inclusion of the @triton.jit decorator, we ensure the model adheres to Triton specifications at the syntactic level.
>
>     c. Experimental Evidence: Our experiments show that the RL stage achieves consistent and significant performance gains over the SFT-only baseline (as shown in Tables 1 and 2), confirming that the model has internalized more reliable Triton programming strategies.
>
> 2. Lack of theoretical model for convergence or performance bounds: We acknowledge that the current work lacks a theoretical convergence analysis of the iterative improvement process. The focus of this paper is to build a systematic, end-to-end framework for LLM-driven kernel generation and optimization, and to demonstrate its effectiveness through empirical evidence. The GRPO algorithm we adopted is also intended to enhance training stability and convergence. We believe that exploring theoretical models is an excellent direction for future research.
>
> 3. Limitations on evaluation scope (≤1024 threads and structured workloads): Our evaluation suites, TRITONBENCH and KERNELBENCH, cover core and diverse operator types and shapes in deep learning (e.g., Matrix Multiplication, Softmax, Layer Norm), representing critical performance bottlenecks in industry practice. We used these standard benchmarks to ensure fair and reproducible comparisons with existing LLM baselines. Future work will extend to more complex, larger-scale fusion and multi-kernel operations.
>
> 4. Generalization to complex fused operators or non-Triton environments: Our data collection pipeline already includes PyTorch fused operations (Fuse). Regarding non-Triton environments (such as native CUDA), AUTOTRITON's methodology is specifically designed for the Triton DSL, leveraging its high-level abstraction features. Generalizing this method to low-level CUDA requires a different environment and data generation mechanism. While theoretically transferable as our pipeline is domain-agnostic, this is beyond the scope of this paper.
>
> 5. Comparison with traditional auto-tuning frameworks (e.g., TVM/Ansor): This is a constructive suggestion. The core contribution of this work is LLM-driven code generation and synthesis, addressing the zero/few-shot programming problem from high-level specifications to executable code. This differs from traditional compilers (like TVM/Ansor/DeepDSL) which primarily optimize existing code through search and transformation. We focus primarily on surpassing existing LLM-driven baselines. Traditional autotuners cannot evaluate automatically; they require manual implementation of baselines for each task in the evaluation set. We believe this should be a consideration for future benchmark construction, but it is outside the scope of this paper.
>
> 6. Dense sections: We accept this feedback and will simplify and clarify the exposition of the data collection pipeline and experimental analysis in the final version to improve readability and scientific rigor.

---

> ### Author Response · Authors · 2025-12-03
> **Part 2/2**
>
> Response to Questions
>
> 1. Integration of performance metrics into LLM feedback: The performance feedback loop described in the your summary differs from our current RL training. As explicitly stated in the paper, due to the lack of sufficient high-quality data, our current training framework does not integrate performance-guided training.
>
>     a. Current Reward: The reward consists only of execution correctness and rule checks (@triton.jit); therefore, performance metrics are not integrated into the training prompts.
>
>     b. Dynamic Prompts: The prompts used for code generation are dynamically adapted. They are based on a template and dynamically populated with the PyTorch kernel specification for each task (including functional description, parameter signatures, and tensor shapes), which are included in the input PyTorch code.
>
> 2. Detection and prioritization of failure types: Fault detection is implemented via the Verifier module and execution tests during data collection and the RL validation stage.
>
>     a. Detection: Any kernel causing compilation failure (syntax/compilation errors), runtime crashes, or numerical results inconsistent with the PyTorch golden standard is marked as a failure.
>
>     b. Prioritization: In RL training, all these error types result in an execution reward of 0. Thus, they have equal priority in the reward function, as functional correctness is the primary goal.
>
> 3. Ensuring performance feedback is interpretable/learning vs. stochastic:
> As mentioned above, we do not currently use performance feedback. However, the model does learn from functional correctness and syntax rewards: Acturally, the RL stage delivers performance gains relative to the SFT model, indicating that the model is learning superior Triton programming patterns and avoiding reward hacking.
>
> 4. Hardware and Environment: We confirm that all experiments were conducted on the same fixed hardware environment and CUDA/Triton versions to ensure fairness and comparability. We have detailed all evaluation protocols and hyperparameters in the experimental section. We will explicitly list specific GPU models and software version information in the final version.

---

### Meta-Review · Area_Chair_Xpuv · 2026-01-15

**Summary:**

The paper proposes AutoTriton, a framework for generating Triton kernels using an 8B parameter LLM fine-tuned via Supervised Fine-Tuning (SFT) and Reinforcement Learning (RL). The authors construct a data pipeline to collect/synthesize kernel data and utilize Group Relative Policy Optimization (GRPO) with rule-based rewards to handle functional correctness and prevent reward hacking (e.g., generating PyTorch wrappers instead of Triton code).

Reviewers generally appreciated the engineering effort in the data pipeline and the demonstration that a smaller model (8B) could compete with larger foundation models. However, significant concerns were raised regarding:

- Benchmarking Integrity: Reliability issues with TritonBench were cited (Reviewer txNq), casting doubt on the reported speedups.
- Reward Formulation: The RL reward loop optimizes for functional correctness and syntax, not runtime performance/latency (Reviewers LZoX, q2RF, txNq).
- Novelty & Baselines: The lack of comparison to traditional auto-tuners (TVM/Ansor) and the similarity to existing AI Engineer paradigms (Reviewers LZoX, q2RF).
- Generalization: Low success rates on out-of-distribution tasks (GitHub data).

**Reviewer Concerns:**

Addressed by Rebuttal:

- Reward Hacking (Reviewer LZoX, 9T7y): The authors effectively clarified how rule-based rewards (forcing @triton.jit) prevent the model from defaulting to PyTorch implementations, a distinct challenge from generic code generation.

- Novelty vs. Agents (Reviewer q2RF): The authors distinguished their work as a fine-tuned model approach rather than a training-free agentic workflow (like AI CUDA Engineer), which is a valid technical distinction.

- Code Example (Reviewer txNq): The authors provided a detailed code snippet demonstrating the model's output, addressing the request for qualitative evidence of reasoning (albeit anecdotal).

Outstanding/Unresolved:

- Performance-Guided Optimization (Reviewers LZoX, q2RF, txNq): A core critique was that AutoTriton does not actually optimize for speed during training, only correctness. The authors acknowledged this is future work due to the cost of data collection. This remains a fundamental limitation for a paper pitching high-performance kernel generation.

- Benchmark Reliability (Reviewer txNq): While the authors stated the results hold despite known TritonBench issues, they did not provide quantitative evidence or new clean runs to definitively alleviate the concern that the improvements are artifacts of a flawed benchmark.

- Comparison to Non-LLM Baselines (Reviewer LZoX): The authors argued traditional compilers (TVM/Ansor) are out of scope because they optimize existing code rather than generate from scratch. While true, these remain the gold standard for performance, and the lack of comparison makes it hard to contextualize the LLM's efficiency.

**Reviewer Scores:**

Reviewer LZoX (Score: 4 -> 4): The reviewer was concerned about the lack of theoretical bounds and integration of performance metrics. The rebuttal confirmed performance metrics are not currently used. The score would likely remain unchanged as the core limitation persists.

Reviewer 9T7y (Score: 6 -> 6): This reviewer was already leaning positive, valuing the efficiency of the 8B model. The rebuttal reinforced the validity of comparing small tuned models against large untuned ones. The score would likely remain stable.

Reviewer q2RF (Score: 2 -> 2): The rebuttal clarified the technical distinction between this work and CUDA agents. However, the reviewer's primary objection—that the reward function is incomplete (ignoring speed)—was acknowledged but not fixed. The score might rise slightly for clarity but remain in the reject range.

Reviewer txNq (Score: 2 -> 2): The reviewer questioned the scientific rigor (TritonBench validity). The authors' assurance that "results hold" without new data is unlikely to fully satisfy a skeptical reviewer, though the provided code example helps.

---

### Decision · Program_Chairs · 2026-01-26

Reject